# A Complex In Vitro Degradation Study on Polydioxanone Biliary Stents during a Clinically Relevant Period with the Focus on Raman Spectroscopy Validation

**DOI:** 10.3390/polym14050938

**Published:** 2022-02-26

**Authors:** Jan Loskot, Daniel Jezbera, Zuzana Olmrová Zmrhalová, Martina Nalezinková, Dino Alferi, Krisztina Lelkes, Petr Voda, Rudolf Andrýs, Alena Myslivcová Fučíková, Tomáš Hosszú, Aleš Bezrouk

**Affiliations:** 1Department of Physics, University of Hradec Králové, Rokitanského 62, 500 03 Hradec Králové, Czech Republic; jan.loskot@uhk.cz (J.L.); daniel.jezbera@uhk.cz (D.J.); 2Center of Materials and Nanotechnologies, Faculty of Chemical Technology, University of Pardubice, Studentska 95, 530 02 Pardubice, Czech Republic; zuzana.olmrovazmrhalova@upce.cz; 3Department of Biology, University of Hradec Králové, Rokitanského 62, 500 03 Hradec Králové, Czech Republic; martina.nalezinkova@uhk.cz (M.N.); alena.fucikova@uhk.cz (A.M.F.); 4Department of Medical Biophysics, Faculty of Medicine in Hradec Králové, Charles University, Šimkova 870, 500 03 Hradec Králové, Czech Republic; alferid@lfhk.cuni.cz (D.A.); lelkesk@lfhk.cuni.cz (K.L.); vodap@lfhk.cuni.cz (P.V.); 5Department of Chemistry, University of Hradec Králové, Rokitanského 62, 500 03 Hradec Králové, Czech Republic; rudolf.andrys@uhk.cz; 6Department of Neurosurgery, Faculty of Medicine in Hradec Králové, Charles University, Sokolská 581, 500 05 Hradec Králové, Czech Republic; tomas.hosszu@fnhk.cz; 7Department of Neurosurgery, University Hospital Hradec Králové, Sokolská 581, 500 05 Hradec Králové, Czech Republic

**Keywords:** biliary stent, polydioxanone, degradation, Raman spectroscopy, tensile strength, Young’s modulus, differential scanning calorimetry

## Abstract

Biodegradable biliary stents are promising treatments for biliary benign stenoses. One of the materials considered for their production is polydioxanone (PPDX), which could exhibit a suitable degradation time for use in biodegradable stents. Proper material degradation characteristics, such as sufficient stiffness and disintegration resistance maintained for a clinically relevant period, are necessary to ensure stent safety and efficacy. The hydrolytic degradation of commercially available polydioxanone biliary stents (ELLA-CS, Hradec Králové, Czech Republic) in phosphate-buffered saline (PBS) was studied. During 9 weeks of degradation, structural, physical, and surface changes were monitored using Raman spectroscopy, differential scanning calorimetry, scanning electron microscopy, and tensile and torsion tests. It was found that the changes in mechanical properties are related to the increase in the ratio of amorphous to crystalline phase, the so-called amorphicity. Monitoring the amorphicity using Raman spectroscopy has proven to be an appropriate method to assess polydioxanone biliary stent degradation. At the 1732 cm^−1^ Raman peak, the normalized shoulder area is less than 9 cm^−1^ which indicates stent disintegration. The stent disintegration started after 9 weeks of degradation in PBS, which agrees with previous in vitro studies on polydioxanone materials as well as with in vivo studies on polydioxanone biliary stents.

## 1. Introduction

Stents are implantable medical devices that are used to keep open obstructed hollow organs [1,2]. Today, a large number of different stent designs with different parameters are available (mechanical properties of the material, radial and expansion force, stent mesh design, etc.) [3,4]. Non-degradable stents can be made of a variety of materials. These materials include stainless steel, nitinol, and polymeric materials (e.g., polyester) [5,6]. At present, there is an effort to use biodegradable materials when it is preferable for the outcome. The use of stents made of these materials, among other benefits, eliminates the need for further invasive procedures [7,8,9]. For clinical use of biodegradable stents, the stent must be able to support the tissue for a defined period and subsequently degrade at an appropriate rate [10]. For this reason, it is necessary to sufficiently describe the degradation process of these materials. Biodegradable materials suitable for stent manufacturing include polymers and metals (e.g., magnesium alloys) [11,12,13]. Polymers are more appropriate for this application because they have a longer degradation time than magnesium alloys [14]. Aliphatic polyesters made of lactide, glycolide, ε-caprolactone, and *p*-dioxanone are frequently used [9,15,16,17]. From the above, the biodegradable polymers contain a hydrolysable bond. Hydrolysis with the contribution of enzymatic activity plays a key role in the degradation of these materials. During hydrolysis, the polymer chains are cleaved [9,18].

Metallic stents are not widely used for the treatment of biliary benign stenoses. Although covered metallic stents are known as retrievable, their use is accompanied by complications with their insertion, migration, and extraction [19,20,21]. To overcome these disadvantages, researchers have recently investigated the potential of self-expandable biodegradable stents for the treatment of biliary benign stenoses [11,22,23]. One of the materials used for biodegradable biliary stents is poly(*p*-dioxanone) (polydioxanone, PPDX). The clinical effects of these stents were investigated by Mauri et al. [22] and De Gregorio et al. [24], and the results of these studies proved the safety and effectiveness of PPDX biliary stents. However, the effect of the degradation process on the material and mechanical properties of PPDX biliary stents is yet to be investigated.

Stents made of polydioxanone can be used in a variety of hollow organs, such as the esophagus [25,26,27,28,29], intestine [30], trachea [31,32,33,34,35], and bile ducts [22,24,36,37,38]. Rejchrt et al. reported that a PPDX stent placed in a gastrointestinal tract maintained its properties (radial force and integrity) for 6–8 weeks, and the stent disintegration occurred after 11–12 weeks. The observed degradation time is sufficient, and therefore the material is suitable for clinical application [39]. 

The manufacturing process of absorbable polydioxanone sutures was originally patented in 1977 [40]. Polydioxanone is a semi-crystalline aliphatic polyester [41,42]. The structure of its crystalline part corresponds to an orthorhombic lattice (P2_1_2_1_2_1_) [43,44]. PPDX hydrolytic breakdown is determined by the crystallinity degree, as hydrolysis occurs preferentially in the amorphous parts of the material [45,46,47]. It has been demonstrated that the hydrolysis medium accesses amorphous regions faster than crystalline regions; once the amorphous portions have been eliminated by hydrolysis, the crystalline regions are affected by the second step of degradation [45]. Several mechanical properties of intravascular [48] and esophageal [49] polydioxanone stents have been investigated. This included measurements of Young’s modulus and shear modulus, fiber elongation, the radial force of stent, and stent force relaxation [48,49,50]. As mentioned above, polyesters (and thus PPDX) degrade by hydrolysis reaction. During degradation, the ratio of crystalline and amorphous phases changes, resulting in a change in mechanical properties [44]. At first, the amorphous phase content decreases [45]. This causes an increase in the radial force of the stent in the initial phase of degradation [48]. Lin et al. observed that 14–28 days of hydrolysis of PPDX sutures in phosphate-buffered saline (PBS) resulted in the loss of roughly 67% of tensile strength, followed by only a low loss of tensile strength between the 42nd and 60th day of degradation. Other degradation phenomena include surface changes, change of thermal properties, and dye leaching (if present) [45].

The vibrational spectra of diverse substances can be studied using Raman spectroscopy (RS), which provides comprehensive information about the bonds between individual atoms in molecules as well as about the material structure. Because RS is usually sensitive to such compounds, it can be useful for polymer analysis. Aside from chemical composition, RS can be used to investigate crystalline qualities (such as crystallite orientation), temperature and mechanical effects on polymers, and also polymer breakdown [51,52,53,54,55]. Raman spectroscopy is a method sensitive enough to track the changes of dye content in a biodegradable stent as it degrades. A previous study showed that during the first 16 weeks of degradation, the PPDX material crystallinity was increasing, indicating that the amorphous phase was preferentially degraded [56].

In this study, Raman spectroscopy measurements of PPDX monofilaments served as a validation of the novel method to evaluate PPDX degradation used by Loskot et al. [56]. Raman measurements were supported by differential scanning calorimetry (DSC). One of the goals of this study was to verify the manufacturer’s claim that the biliary stent integrity and radial force are maintained for 6 to 8 weeks after implantation. Therefore, a degradation period of 9 weeks was chosen. To the best of our knowledge, there is no study comparing the results of Raman spectroscopy with the mechanical properties of a PPDX monofilament. Therefore, this study compares the Raman spectroscopy results with mechanical properties of PPDX monofilaments degraded over 9 weeks in PBS. 

## 2. Materials and Methods

### 2.1. Stent Degradation

The research samples were biliary biodegradable stents (Figure 1a) manufactured by ELLA-CS (Hradec Králové, Czech Republic). These stents were braided from 0.39 mm thick monofilament made of poly(*p*-dioxanone) with added 1-hydroxy-4-(*p*-tolylamino)anthracene-9,10-dione (also referred to as Solvent Violet 13, content < 0.1 wt%). The stent length is 80 mm and its diameter is 8 mm. The mass of an intact stent is 0.32 g.

A total of 32 stents were analyzed, and degradation periods of 0, 3, 6, and 9 weeks were chosen. 8 stents were incubated for each of these periods. Accordingly, we refer to stents after these degradation periods as test groups 0W (before degradation), 3W (degraded for 3 weeks), 6W (degraded for 6 weeks), and 9W (degraded for 9 weeks). Each stent was incubated separately in a sealed tube filled with PBS. The tubes were shaken all the time in a circular vibrating shaker Vibramax 100 (Heidolph, Schwabach, Germany) placed in a thermostatic cabinet Liebherr FK 3640 (Liebherr Haushaltsgeräte, Ochsenhausen, Germany) at a constant temperature of 37 °C. Each stent was cut with scissors into smaller samples (four 70 mm long pieces, three 20 mm long pieces—Figure 1b) under aseptic conditions in a safety cabinet Safemate EVO 1.2 (BioAir, Camden, NJ, USA) after its degradation period.

### 2.2. Thermal Analysis

The moisture content of PPDX stents after hydrolytic degradation was determined by thermogravimetric analysis (TGA) using STA 449 F5 Jupiter (Netzsch, Selb, Germany). The experiments were performed in an aluminum crucible equipped with a special lid with a small hole to minimize moisture loss before the start of the measurement. The weight of samples was kept between 10 and 13 mg. Samples were heated from 30 °C to 150 °C at a heating rate of 5 °C min^−1^ under a dynamic nitrogen atmosphere (flow rate 50 mL min^−1^).

The crystallinity and thermal stability of the PPDX stents were examined using a Q2000 heat flow differential scanning calorimeter (TA Instruments, New Castle, DE, USA) equipped with an autosampler, RCS90 cooling accessory, and T-zero technology. Dry nitrogen was used as purge gas at a flow rate of 50 mL min^−1^. The sample masses were approximately 8 mg. DSC measurements were done in the following steps: (1) isotherm for 5 min at 25 °C, (2) heating scan from 25 °C to 150 °C to register melting behavior of ”as received” polymer, (3) isotherm for 5 min at 150 °C to erase the thermal history, (4) cooling scan from 150 °C to −30 °C, (5) isotherm for 5 min at −30 °C, and finally (6) heating scan from −30 °C to 150 °C. Steps (4) and (6) were used to characterize the glass transition and crystallization behavior after the elimination of previous thermal history. The scanning rate was 5 °C min^−1^. 

The crystallinity of PPDX was calculated according to the following equation:$$X_{c}=\frac{\Delta H_{m1}}{\Delta H_{m100\%}}\times 100\%$$
in which Δ*H_m_*_1_ is the enthalpy of melting of the PPDX during the first heating scan and Δ*H_m_*_100%_ is the enthalpy of melting for 100% crystalline PPDX, which was taken as 141.18 J g^−1^ [41].

### 2.3. Scanning Electron Microscopy

Morphological changes in the stents induced by the degradation process were observed by a scanning electron microscope (SEM) FlexSEM 1000 (Hitachi, Tokyo, Japan). The surface of the stents before degradation and after each degradation period was imaged with 1500× magnification. The SEM was working in secondary electrons mode and the accelerating voltage used was 15 kV. Before the measurement, all samples were coated with a 7 nm thick golden layer using a sputter coater EM ACE200 (Leica, Wetzlar, Germany) to increase their electrical and thermal conductivity.

### 2.4. Infrared Spectroscopy

Infrared spectra of the stents before degradation and after each degradation period were obtained by ALPHA II Fourier transform infrared (FT-IR) spectrometer equipped with a platinum-ATR-sampling module (diamond) (Bruker, Billerica, MA, USA). The measurements were performed in triplicate and averaged. The FT-IR spectra were taken in the range from 400 to 4000 cm^−1^, the acquisition time of each spectrum being 70 s. The aim was to check whether some chemical changes in the material occurred during the degradation process.

### 2.5. Raman Spectroscopy

Raman spectra of the stents before degradation and after each degradation period were measured by XploRA PLUS (Horiba, Kyoto, Japan) dispersive Raman micro-spectrometer. The excitation laser wavelength was 785 nm, which was found to provide a signal of higher quality compared to the 638 nm laser. The laser power was 10 mW, ensuring that the samples were not thermally damaged. The signal was acquired using a 10× objective and an 1800 g mm^−1^ (grooves per millimeter) grating. 

Raman spectra were taken from the top surface of each sample in a regular 8 × 4 grid. Thus, 32 spectra were obtained from each stent. The measurement arrangement is shown in Figure 2. To suppress the influence of sample orientation on the spectra, all the stent monofilaments were positioned parallel to the polarization plane of the excitation laser. The acquisition of each spectrum took approximately 3 min 26 s.

The spectral range measured was 1550–1860 cm^−1^. In this spectral region, the hydrolytic degradation of PPDX is clearly noticeable, as we described in [56]. In the mentioned study, we revealed that the shoulder of the 1732 cm^−1^ peak (C=O stretching vibrations in the ester carbonyl group [57]) decreases during the degradation, enabling a quantitative evaluation of the progression of PPDX hydrolytic degradation. Therefore, we also focused on this region in the current study.

In addition to this, wider-range Raman spectra (150–2100 cm^−1^) were taken from each sample to check whether some other changes in the material spectrum occur during the degradation process. The same laser wavelength as before and a 1200 g mm^−1^ grating was used. The acquisition time of each spectrum was about 3 min 35 s. The measurements were performed in triplicate and averaged.

### 2.6. Mechanical Properties

#### 2.6.1. Young’s Modulus

In each test group (0W, 3W, 6W, and 9W), we measured Young’s moduli of 8 stent monofilament samples using ordinary tensile tests performed on an Instron 3343 Single Column Testing System (Instron, Norwood, MA, USA) with a 1 kN force transducer and Instron’s flat pneumatic grips. We performed the measurements at a laboratory temperature of (25 ± 1) °C. We also monitored relative humidity for extreme fluctuations, i.e., a difference of more than 10%, due to polydioxanone’s natural hydrophilicity that could potentially affect the measured data. We used a standard measurement procedure that has already been reported in several studies [49,58,59]. We gathered the force-extension data, which we subsequently transformed to standard stress–strain curves using the sample monofilament diameter *d* and the original effective length of the sample *L_TE_*. We performed 20 measurements of *L_TE_* using the Extol 3426 caliper (Madal Bal a. s., Zlín, Czech Republic). We determined Young’s modulus as the slope of the greatest pseudolinear part of the stress–strain curve.

#### 2.6.2. Tensile Strength

Tensile strength is an important material property related to the process of degradation, showing the stent resistance to disintegration. This property has a major clinical impact. From the tensile data, we analyzed the tensile strength of samples and its relation to the degradation process. To express the sample tensile strength, we used the maximum force measured before the rupture of the sample. We compared the data in consecutive degradation steps (i.e., non-degraded vs. 3 weeks of degradation, 3 vs. 6 weeks, and 6 vs. 9 weeks) to find a significant tensile strength decrease. We also compared the maximum forces to a limit force *F_L_* to assess the stent disintegration resistance. For the tested biliary stents, the axial force necessary to completely compress or extend the stent is never greater than *F_MAX_* = 10 N. Taking into account possible local overloads of the stent monofilament given (e.g., caused by the specific geometry of the treated tissue), we set a risk factor to *r* = 10. Considering the number of stent branches *n* = 16 and the risk factor *r*, the limit force should be greater than:$$F_{L}>F_{MAX}\frac{r}{n}=6.25\;N$$

Therefore, we chose *F_L_* = 7 N as a clinically relevant limit force for a single stent monofilament.

#### 2.6.3. Shear Modulus and Rigidity

For shear modulus and shear stiffness measurements, we used the same procedure and device as thoroughly described, including instructional pictures and videos, by Bezrouk et al. [49]. The measuring device was attached to the Instron 3343 equipped with a 1 kN force transducer. We performed the measurements at a laboratory temperature of (25 ± 1) °C and monitored relative humidity for extreme fluctuations over 10%, due to polydioxanone’s natural hydrophilicity that could potentially affect the measured data.

In this method, a specimen (a piece of polydioxanone stent monofilament) is placed between two coaxial micro chucks, fixed, and twisted (Figure 3).

From the force-extension data profile, the total system rigidity ${\tilde{K}}_{T}$ (measurement with a specimen) and the device rigidity ${\tilde{K}}_{D}$ (20 measurements taken for the device without a specimen) were determined. Then, the specimen rigidity is ${\tilde{K}}_{S}={\tilde{K}}_{T}-{\tilde{K}}_{D}$. The shear modulus of a polydioxanone monofilament was eventually calculated as follows:$$G\;=\frac{8{\bar{D}}^{2}{\bar{L}}_{W}\left({\tilde{K}}_{T}-{\tilde{K}}_{D}\right)}{\pi {\bar{d}}^{4}}$$
where ${\bar{L}}_{W}$ is the mean specimen length (20 measurements using the Extol 3426 caliper (Madal Bal a. s., Zlín, Czech Republic)), $\bar{D}$ is the mean diameter of the pulley of the measuring device obtained from 20 measurements using the Extol 3426 calipers, and $\bar{d}$ is the mean stent monofilament diameter (10 measurements using the Extol 3426).

We also tested whether the measured shear rigidities are higher than 0 N, due to the minuscule force values measured during this test, which are at the limit of the measuring device capabilities.

### 2.7. Statistics

The acquired Raman spectra were further statistically evaluated using MATLAB R2020a software (2020, MathWorks, Natick, MA, USA) and NCSS 10 statistical software (2015, NCSS, LLC., Kaysville, UT, USA, Available online: ncss.com/software/ncss (accessed on 22 February 2022)). 

The need for the statistical evaluation stems from the fact that the 1732 cm^−1^ peak with its shoulder is relatively low in intensity, resulting in a deterioration of the signal-to-noise ratio. For the samples before degradation and after each degradation period, the area under the 1732 cm^−1^ peak shoulder (between 1736 and 1757 cm^−1^) was calculated and normalized to the height of the 1732 cm^−1^ peak. The first step was to remove baselines from the spectra to ensure accurate quantification of the 1732 cm^−1^ peak height and its shoulder area. Subsequently, the normalized shoulder areas were determined for all the 32 spectra of the particular stent, and the results were averaged. The normalization was done by dividing the shoulder area by the 1732 cm^−1^ peak height. The area calculation procedure can therefore be considered as a relative method. This procedure is described in detail in Appendix A.

Finally, these mean values of different samples were compared to find their dependency on the degradation time.

We used the D’Agostino omnibus test to test the normality of the data distribution. The data from normally distributed populations with more than 10 results in a test group were described using the mean and standard deviation of the sample ($\bar{X}$ ± SD) while the other data were described using the median and the first and third quartiles of $\tilde{X}$ (1st Q, 3rd Q). Due to the limited number of samples in some test groups, we opted to use the Wilcoxon signed-rank test. To adjust for multiple comparisons and keep the familywise α at 0.05, the Bonferroni correction was used. The resulting α for a single comparison was 0.0167.

## 3. Results and Discussion

### 3.1. Stent Degradation

The pH values of PBS solution (Table 1) after the stent degradation were measured. The pH values showed a decreasing trend, which is probably caused by the acidic degradation products of PPDX. The biochemical degradation pathway of PPDX is described by Ciconne et al. [60]. Investigation of this phenomenon was not the focus of this study, and there is a need for further research. 

### 3.2. Thermal Analysis

Two methods of thermal analysis, thermogravimetric analysis, and differential scanning calorimetry were used to study the thermal stability of a non-degraded stent and stents after 3, 6, and 9 weeks of hydrolytic degradation.

#### 3.2.1. Thermogravimetric Analysis

The water absorbed by the sample during the degradation was detected in the 30–150 °C temperature range by TGA. The weight loss of all the measured samples ranged from 0.2 to 0.5%. No trend occurs in the data. Thus, the earlier assumption that water content in the material increases during degradation was not confirmed. 

#### 3.2.2. Thermal Behavior and Determination of Crystallinity

The effect of degradation period on thermal behavior of PPDX stents was studied by DSC. As the degradation progressed, changes in the shape and area of the melting and crystallization peaks and their slight temperature shifts were observed. Figure 4 shows a comparison of dynamic DSC curves of a non-degraded sample and a 9-weeks-degraded sample during the first heating, cooling, and the second heating rate (All DSC measurement protocols are available in Appendix A). A substantial difference between the non-degraded and degraded samples is already evident in the endothermic melting peaks during the first heating. The melting peak of the non-degraded sample shows two maxima in the 99–107 °C range. With increasing degradation period, the position of the peak maximum shifts to higher temperatures (up to 110 °C), and at the same time the peak split gradually disappears, and the peak is more intense and narrower.

During cooling from 150 °C to −30 °C, the crystallization process from melt occurred. Another two exothermic crystallization peaks and one endothermic melting peak can be seen during the second heating DSC scan. In contrast to the melting temperature during the first heating, the temperature of these processes shifts only slightly with increasing degradation period, but their enthalpy changes substantially. The values of their enthalpies as a function of the degradation time are plotted in Figure 5. There is a considerable increase in the enthalpy of crystallization during cooling, which changes from (18.4 ± 2.4) J g^−1^ for the non-degraded sample to (46.7 ± 4.8) J g^−1^ for the 9-weeks-degraded sample. It is probably because the stent morphology is disturbed during hydrolysis, leading to a higher tendency to crystallize from the melt. The morphology changes could also be related to the change in the shape of the melting peak and its temperature shift during the first heating. Conversely, during subsequent second heating, the peak corresponding to cold crystallization shrinks, and its enthalpy, decreases from (16.8 ± 1.9) J g^−1^ for the non-degraded sample to (2.2 ± 1.2) J g^−1^ for the 9-weeks-degraded sample. 

The enthalpy of melting during the first heating was taken to calculate the value of crystallinity of all PPDX stents. The results are summarized in Table 2. The crystallinity increases with the degradation period, which is in good agreement with previously published works [56,61,62,63]. In the beginning, the increase in the proportion of the crystalline phase was small (around 2%). Between the 6th and the 9th week of degradation, the change was higher (around 4%). All the crystallinity increases were statistically significant (non-degraded to 3-weeks-degraded samples: *p* = 0.016; 3-weeks-degraded to 6-weeks-degraded samples: *p* = 0.012; 6-weeks-degraded to 9-weeks-degraded samples: *p* = 0.016). The increase in crystallinity can have two reasons. The first possible reason is that, predominantly, the amorphous phase elutes during hydrolysis, while the crystalline phase is more resistant [61]. The second reason is that the hydrolysis is performed at body temperature (37 °C), which is well above the glass transition temperature ((−10.7 ± 0.1) °C for a non-degraded sample) but below the melting temperature. In this temperature range, the thermal annealing process occurs simultaneously with hydrolytic degradation, leading to further gradual crystallization of the polymer. Similar results were previously reported by Zhang et al. [62].

### 3.3. Scanning Electron Microscopy

SEM observations revealed that stents before degradation have fairly smooth, non-damaged surfaces (Figure 6a). After 3 weeks of degradation, small fibrils attached to the surface could be seen (Figure 6b). 6-weeks-degraded samples looked similarly, again with fibrils visible on the surface (Figure 6c). Finally, after 9 weeks of degradation, the surface was rougher in some areas, and small pieces of material were peeling off (Figure 6d). No crazes or cracks were observed even after this degradation period. It is not very surprising, as in the observation reported by Loskot, et al. [56], esophageal PPDX stents after 8 weeks of degradation in PBS only rarely showed tiny crazes. This slight discrepancy in the surface deterioration speed can be explained by the fact that different types of stents were analyzed in these studies. 

### 3.4. Infrared Spectroscopy

FT-IR spectra of the stents (Figure 7) were very similar to each other, almost unaffected by the degradation process. Only little changes can be assumed in the region between 1500 and 1650 cm^−1^. As suggested in [56], these changes could be associated with the loss of dye in the material and with the formation of carbonyl anions caused by the hydrolysis of ester bonds. The mutual similarity of all the spectra indicates that no other functional groups were formed during the degradation, which is consistent with [56]. All the measured FT-IR spectra are provided in Appendix A.

### 3.5. Raman Spectroscopy

Figure 8 shows wide-range Raman spectra (150–2100 cm^−1^) of the stents after all degradation periods. The 483, 1242, 1403, 1610, and 1638 cm^−1^ peaks can be attributed to the dye, and the other peaks belong to PPDX itself [56]. As can be seen in Figure 8, except for the 1732 cm^−1^ peak shoulder, only the peaks of the dye changed (decreased) during the degradation. It means that the dye is gradually eluted from the stents during the degradation. The “dye” peaks, however, did not disappear completely during the 9 weeks of degradation, showing that some dye still remains in the material. All the measured Raman spectra are provided in Appendix A.

The progressing degradation induced the lessening of the 1732 cm^−1^ peak shoulder, as shown in Figure 9. The corresponding normalized areas under this shoulder are given in Table 3. All these decreases in the 1732 cm^−1^ peak normalized shoulder area are statistically significant (*p* < 0.001).

### 3.6. Mechanical Properties

#### 3.6.1. Young’s Modulus

The determined Young’s moduli of polydioxanone monofilaments taken from non-degraded stents and stents after 3, 6, and 9 weeks of degradation in PBS are shown in Figure 10. Young’s modulus of PPDX monofilaments taken from non-degraded stents is $\tilde{E}$ = 965 (948, 1042) MPa. A statistically significant decrease in Young’s modulus (*p* < 0.001) was observed only for 9-weeks-degraded stents (All results of Young’s modulus measurements are available in Appendix A).

The determined Young’s modulus of the non-degraded PPDX monofilament is $\tilde{E}$ = 965 (948, 1042) MPa, which is in good agreement with previously published values for PPDX monofilaments [49,58]. Nevertheless, the material processing during the stent production should be considered as it may significantly affect the resulting properties of the stent material.

The material stiffness is maintained during the first 6 weeks of degradation. With respect to the TGA results (Section 3.2.1), it is probably caused by a mechanism different from the sample hydration considered in some earlier studies [49,56,64]. It seems that the initial maintenance of Young’s modulus may result from the increase in the proportion of the crystalline phase in the material.

The decrease in Young’s modulus of the PPDX monofilament was significant only in 9-weeks-degraded samples. The same was reported in some previous in vitro studies investigating the chemical degradation of polydioxanone [65,66]. Although Young’s modulus also shows changes between the other phases of material degradation, these changes are neither statistically nor clinically significant and can be attributed to higher data variability caused by the limited number of samples tested.

#### 3.6.2. Tensile Strength

Figure 11 shows tensile strengths of PPDX monofilaments taken from non-degraded stents and stents after 3, 6, and 9 weeks of degradation. We observed a statistically significant decrease in the tensile strength of 6-weeks-degraded monofilaments compared to 3-weeks-degraded monofilaments (*p* < 0.001) as well as for 9-weeks-degraded monofilaments compared to 6-weeks-degraded monofilaments (*p* < 0.001) (All results of tensile strength measurements are available in Appendix A.

The maximum forces measured during the test were significantly higher than the limit force *F_L_* = 7 N for the non-degraded stent monofilaments (*p* = 0.007) and for the stent monofilaments after 3 (*p* = 0.007) and 6 (*p* = 0.007) weeks of degradation.

The tensile strength results show a statistically significant decrease after all degradation periods, which agrees well with the increase in the material crystallinity determined from the Raman spectra and DSC analyses. It could be due to the rising fragility of the samples caused by the increase in the proportion of the crystalline phase in the polydioxanone material during the degradation process. These findings are clinically important as the tensile strength of stent material is closely related to the stent disintegration resistance. However, only a decrease in tensile strength below a certain critical (or limit) value is clinically significant. Considering the maximum possible forces in a clinically relevant situation, we estimated the limit force value for a single stent monofilament wire *F_L_* = 7 N. Only PPDX monofilaments from the 9-weeks-degraded stents showed tensile strength below the limit force of 7 N. These findings are consistent not only with the manufacturer’s declared stent durability but also with previous in vitro studies of polydioxanone materials and stents [49,56,65,66,67,68,69] stating that the radial force of the stent, and hence the stent integrity, is maintained for 6–8 weeks after its implantation.

#### 3.6.3. Shear Modulus and Rigidity

Figure 12 shows shear rigidities of PPDX monofilaments taken from non-degraded stents and stents after 3, 6, and 9 weeks of degradation. The determined rigidities were significantly higher than 0 N m^−1^ after all degradation periods (non-degraded samples: *p* = 0.007; 3-weeks-degraded samples: *p* = 0.021; 6-weeks-degraded samples: *p* = 0.007; 9-weeks-degraded samples: *p* = 0.040).

The shear modulus of PPDX monofilaments taken from non-degraded stents is $\tilde{G}$ = 349 (238, 746) MPa. A significant shear rigidity decrease was observed only for 9-weeks-degraded samples (*p* = 0.010) (All results of shear modulus and rigidity measurements are available in Appendix A).

To investigate the polydioxanone stent material to a greater extent and provide some information interesting for further stent development, we also studied the shear modulus and rigidity of the polydioxanone material. By comparing the shear rigidity values against the value of 0 N, we verified that the measured values are non-random and, therefore, caused by the shear rigidity of the tested sample. However, their reliability is very low. The determined shear modulus of the non-degraded PPDX monofilament is $\tilde{G}$ = 349 (238, 746) MPa, which agrees with the previously published value [49]. However, the wide interquartile range shows that the reliability of this result is also very low. The comparisons of the shear rigidities revealed only a single significant decrease between the 6-weeks-degraded and 9-weeks-degraded samples, which is most probably only a random result arising from the irregularity of the data for the 6-weeks-degraded samples. The low reliability of the shear modulus and rigidity results can be partly attributed to the high data variability due to the limited number of samples tested and partly to the limited capabilities of the measuring device to detect such low force values.

### 3.7. Comparison of DSC, RS, and Tensile Strength Data 

Figure 13 shows the changes in amorphicity, tensile strength, and the normalized area under the 1732 cm^−1^ Raman peak shoulder, occurring during the first 9 weeks of hydrolytic degradation of the stents. The values presented in this graph are the results taken from the corresponding Section 3.2, Section 3.5 and Section 3.6.2. By the term “amorphicity” we mean the ratio of the amorphous phase in the material, i.e., a complement to the crystallinity (expressed in %). To be more easily comparable, the values presented in the plot are normalized so that the initial values (before the degradation started) are set to 100%.

The relation between the amorphicity and the normalized shoulder area is almost the same as in the study by Loskot et al. [56]. The small difference can be due to a different type of stents analyzed in the previous study [56]. Another possible reason is that Raman measurements in the current study were performed with a better spectral resolution.

The data on amorphicity and normalized peak shoulder area correlate well with the tensile strength data. Considering Pearson’s correlation coefficients, the amorphicity shows a medium correlation (Figure 14a) [70] (R = 0.740) and the normalized peak shoulder area shows a very strong correlation (Figure 14b) [70] (R = 0.947) with the tensile strength. Compared with the amorphicity and peak shoulder area, the tensile strength decreases more progressively. At 9 weeks of degradation, the tensile strength dropped below 3.5 N (7% of its initial value), which is less than the limit force of *F_L_* = 7 N. The decrease in the tensile strength of the tested stent monofilament below the limit force *F_L_* means that the tested type of biliary stent begins to disintegrate under clinically relevant conditions. At the same time, the peak shoulder area decreased to 8.86 cm^−1^ (80% of its initial value). Therefore, the normalized shoulder area of less than 9 cm^−1^ can be considered the level of PPDX degradation at which the biliary stent disintegrates.

### 3.8. Strengths and Limitations

The study was focused on the first 9 weeks of hydrolytic degradation of the stents, which is a clinically relevant period.

The Raman measurements in the 1732 cm^−1^ peak region were done using an 1800 g mm^−1^ grating, which allowed high spectral resolution to be achieved. Further, the acquisition of 32 spectra from each stent resulted in noise reduction in the data and obtaining high-quality and reproducible spectra of the 1732 cm^−1^ peak region. 

The used method of Raman measurements was successfully validated by tensile strength tests and DSC analysis.

A limitation of this study is the number of stents analyzed. Unfortunately, the availability of these special stents was limited due to their high cost and low production capacity. Eight stents per degradation period is considered sufficient for statistical processing, however, it would be advisable to use larger test groups to obtain more robust results.

## 4. Conclusions

The DSC, Raman spectroscopy, and tensile strength data identically show a gradual degradation of polydioxanone filaments in PBS. The tensile strength data confirm that the stent disintegration process begins at 9 weeks after the onset of degradation. These findings are consistent with previous in vitro studies on polydioxanone materials and stents and in vivo studies on polydioxanone biliary stents showing that the radial force of the stent and hence the stent integrity is maintained for 6–8 weeks following implantation.

Raman spectroscopy has proven to be a sufficiently robust method to assess polydioxanone degradation. At the 1732 cm^−1^ Raman peak, the normalized shoulder area of less than 9 cm^−1^ indicates the level of PPDX degradation at which the biliary stent disintegrates.

## Figures and Tables

**Figure 1 polymers-14-00938-f001:**
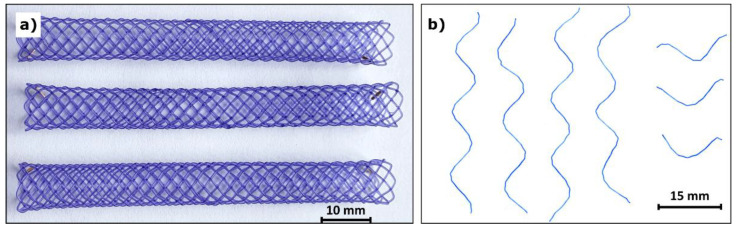
Biodegradable polydioxanone biliary stents (**a**) and an example of monofilament samples (**b**).

**Figure 2 polymers-14-00938-f002:**
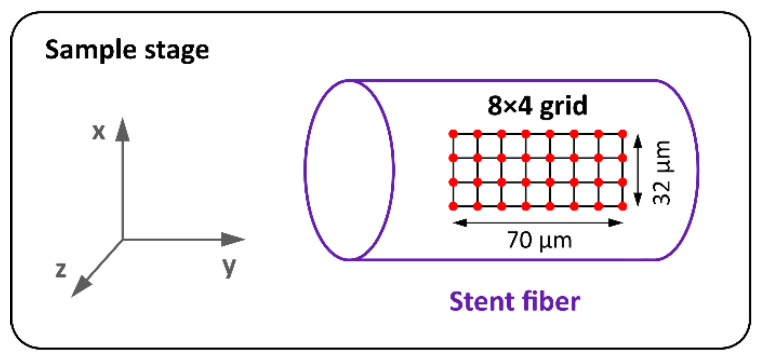
Raman measurements arrangement. The excitation laser beam is directed at the sample along the z (vertical) axis.

**Figure 3 polymers-14-00938-f003:**
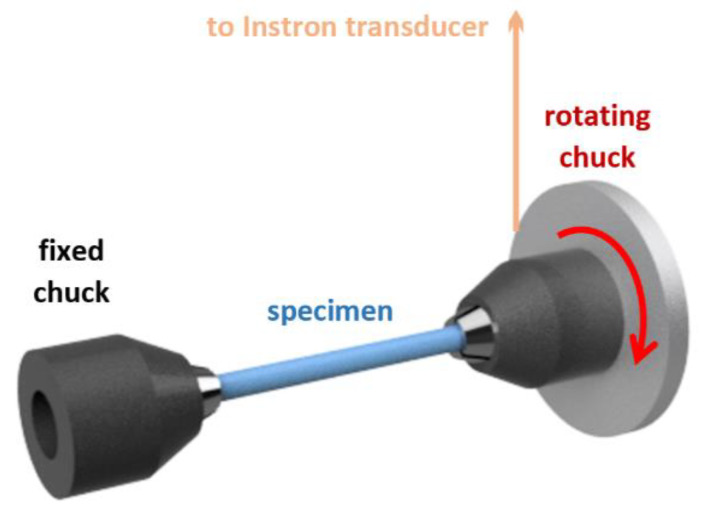
The setup of the measurement device with a fixed specimen.

**Figure 4 polymers-14-00938-f004:**
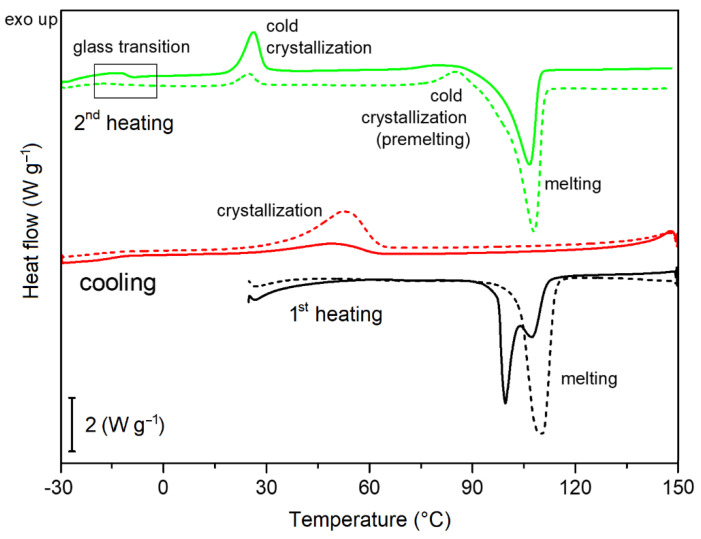
Comparison of differential scanning calorimetry (DSC) curves of a non-degraded (full line) and 9-weeks-degraded stent (dashed line). All runs were performed at the rate of 5 °C min^−1^.

**Figure 5 polymers-14-00938-f005:**
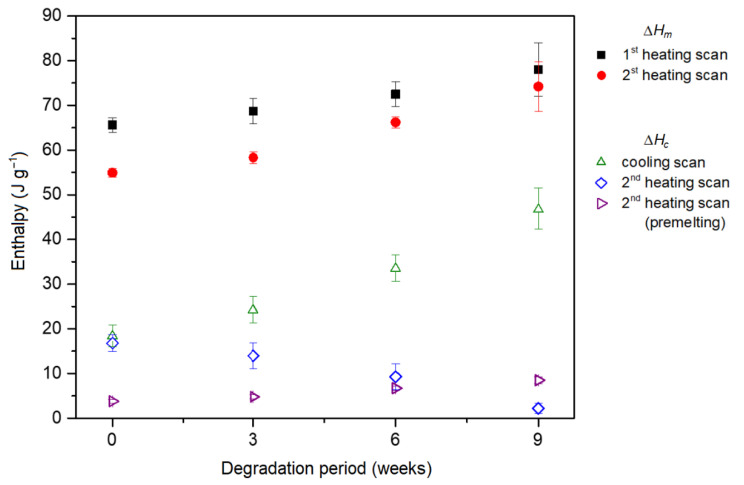
Enthalpy of melting (Δ*H_m_*) and crystallization (Δ*Hc*), determined during DSC measurements of polydioxanone (PPDX) stents, as a function of the degradation period.

**Figure 6 polymers-14-00938-f006:**
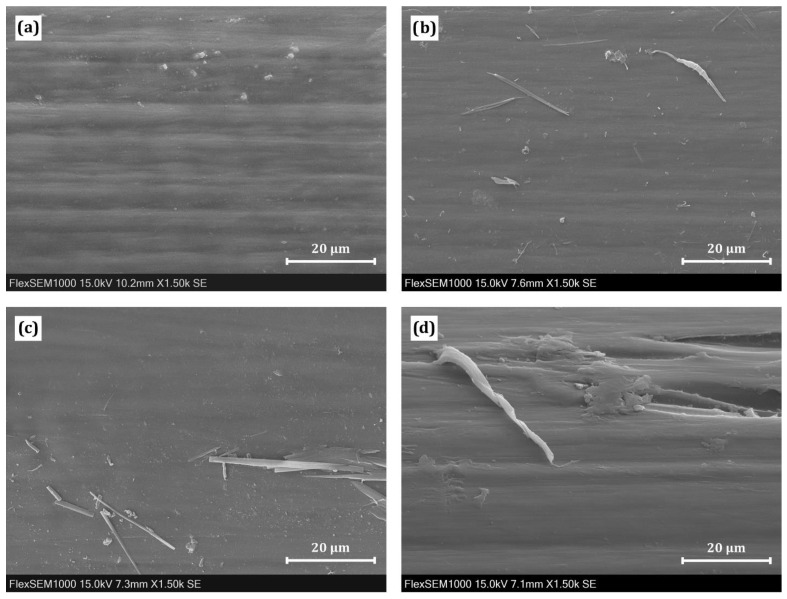
Surface morphology of the stents: (**a**) before degradation; (**b**) after 3 weeks; (**c**) after 6 weeks; and (**d**) after 9 weeks of degradation in phosphate-buffered saline (PBS).

**Figure 7 polymers-14-00938-f007:**
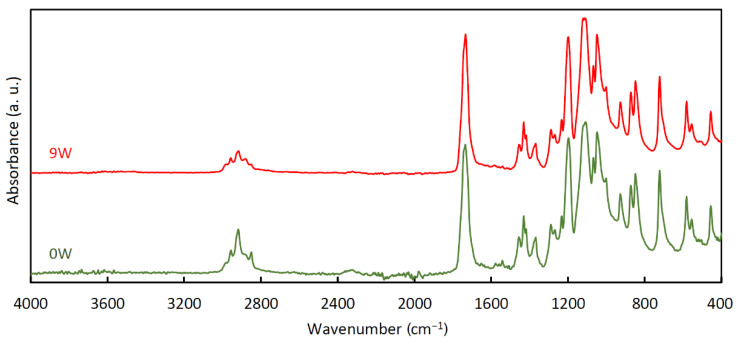
Normalized and averaged Fourier transform infrared (FT-IR) spectra of the stents before degradation (0W) and after 9 weeks of degradation (9W).

**Figure 8 polymers-14-00938-f008:**
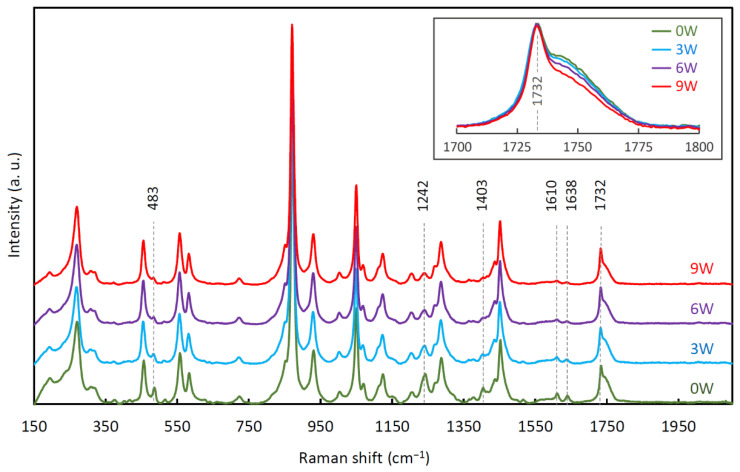
Normalized and averaged Raman spectra of the stents before degradation (0W), after 3 weeks (3W), 6 weeks (6W), and 9 weeks (9W) of degradation. Baselines were removed using LabSpec 6 software. The inset shows a detail of the 1732 cm^−1^ peak and its shoulder.

**Figure 9 polymers-14-00938-f009:**
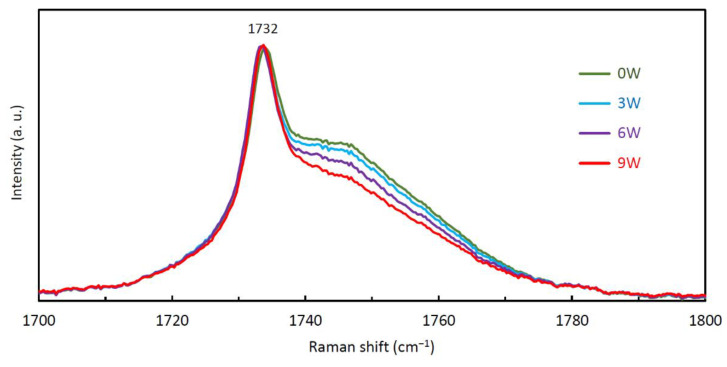
The gradual decrease of the 1732 cm^−1^ peak shoulder. Normalized and averaged Raman spectra of the stents before degradation (0W), after 3 weeks (3W), 6 weeks (6W), and 9 weeks (9W) of degradation.

**Figure 10 polymers-14-00938-f010:**
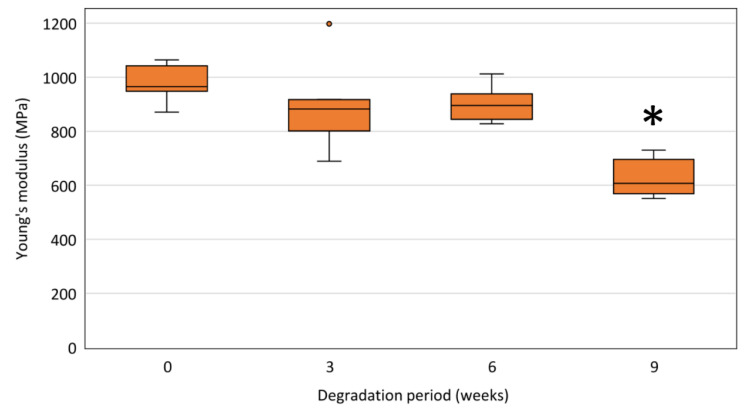
Young’s moduli of stent monofilaments after respective degradation periods. The samples were measured at a laboratory temperature of (25 ± 1) °C. The difference between the maximum and minimum relative humidity values did not exceed 10%. The asterisk (*) indicates a statistically significant decrease of Young’s modulus at the respective time step compared to the previous time step.

**Figure 11 polymers-14-00938-f011:**
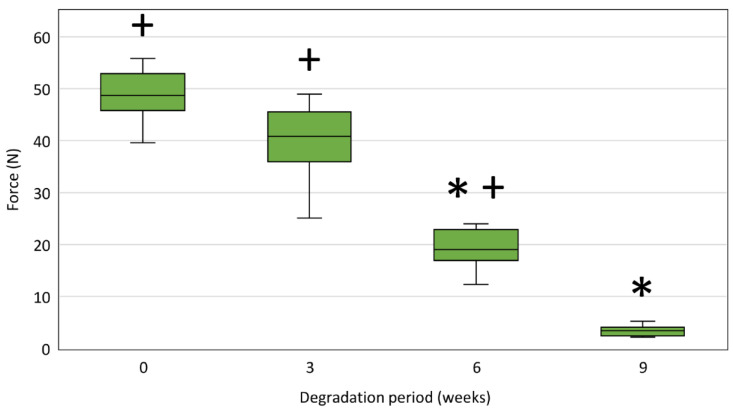
Tensile strengths of stent monofilaments after respective degradation periods expressed as the maximum force measured during the test. The samples were measured at a laboratory temperature of (25 ± 1) °C. The difference between the maximum and minimum relative humidity values did not exceed 10%. The asterisk (*) indicates a statistically significant force decrease at the respective time step compared to the previous time step. The plus sign (+) indicates the maximum forces are significantly higher than the limit force *F_L_* = 7 N.

**Figure 12 polymers-14-00938-f012:**
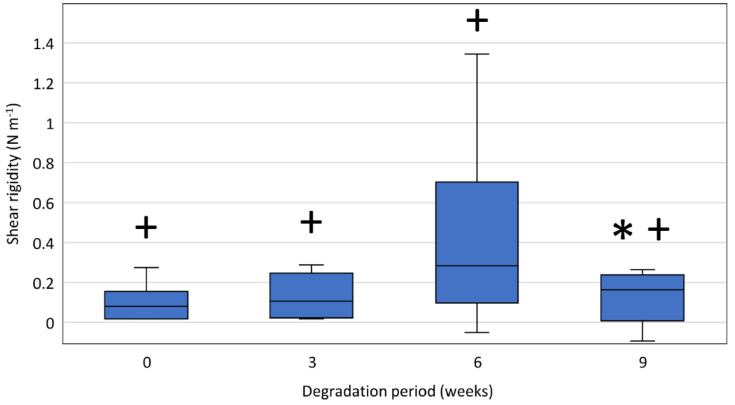
Shear rigidity of stent monofilaments after respective degradation periods. The samples were measured at a laboratory temperature of (25 ± 1) °C. The difference between the maximum and minimum relative humidity values did not exceed 10%. The asterisk (*) indicates a statistically significant shear rigidity decrease at the respective time step compared to the previous time step. The plus sign (+) indicates the shear rigidities significantly higher than 0 N m^−1^.

**Figure 13 polymers-14-00938-f013:**
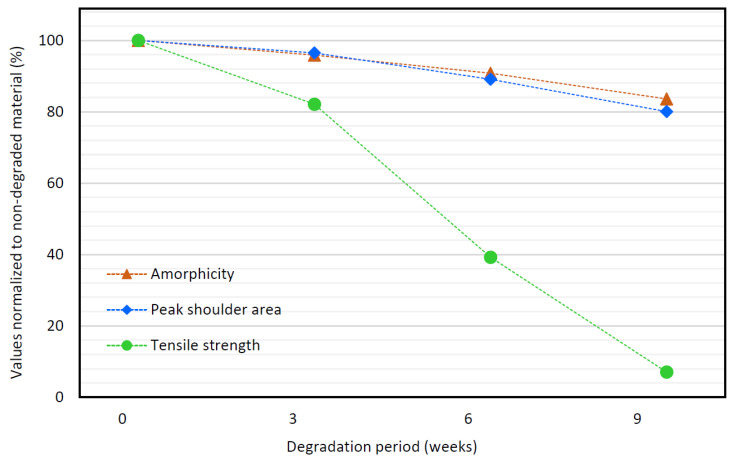
Time course of variables characterizing the hydrolytic degradation. The values are normalized to the values belonging to non-degraded biliary stents, i.e., 100% corresponds to the initial values of amorphicity, 1732 cm^−1^ peak shoulder area, and tensile strength.

**Figure 14 polymers-14-00938-f014:**
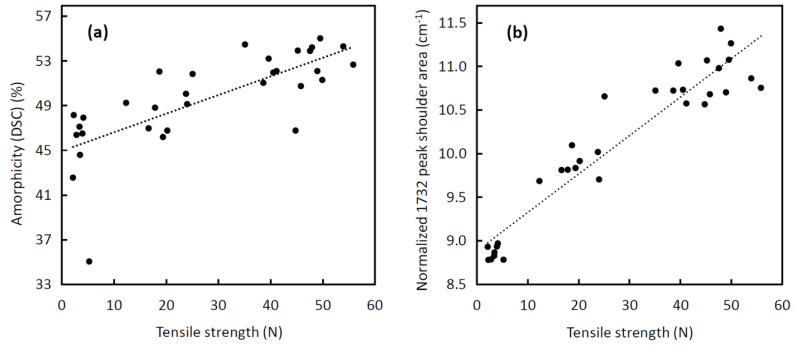
(**a**) Correlation of amorphicity obtained from DSC with the tensile strength; (**b**) correlation of normalized 1732 cm^−1^ peak shoulder area of the Raman spectra with the tensile strength. Both scatter plots show paired data of all 32 stents and the degradation periods of 0, 3, 6, and 9 weeks.

**Table 1 polymers-14-00938-t001:** pH values of PBS solution decreasing with the stent degradation period.

Degradation Period	pH
0 weeks	7.45 ± 0.01
3 weeks	7.29 ± 0.21
6 weeks	7.23 ± 0.25
9 weeks	7.19 ± 0.34

**Table 2 polymers-14-00938-t002:** Crystallinity of the PPDX stents increasing with the degradation period.

Degradation Period	Crystallinity (%)
0 weeks	46.4 ± 1.2
3 weeks	48.6 ± 2.2
6 weeks	51.3 ± 1.9
9 weeks	55.2 ± 4.3

**Table 3 polymers-14-00938-t003:** The dependence of the normalized area under the 1732 cm^−1^ peak shoulder (calculated between 1736 and 1757 cm^−1^) and amorphicity on the degradation period.

Degradation Period	Normalized Area (cm^−1^)	Amorphicity (%)
0 weeks	11.06	53.6 ± 1.2
3 weeks	10.67	51.4 ± 2.2
6 weeks	9.86	48.7 ± 1.9
9 weeks	8.86	44.8 ± 4.3

## Data Availability

The data presented in this study are available in the Appendix A.

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
