# Peer review of "A Complex In Vitro Degradation Study on Polydioxanone Biliary Stents during a Clinically Relevant Period with the Focus on Raman Spectroscopy Validation"

_polymers, 2022, doi:10.3390/polym14050938_

Round 1

Reviewer 1 Report

This topic is interesting and is suitable to be published in Polymers. There are some comments for revision:

Point 1: The manuscript should be carefully revised, some typos could be observed in the introduction, methods and discussion.

Point 2: Some other changed properties of stents such as weight or pH changes should be needed.

Point 3: Some studies of biodegradable metal and polymer for stents should be cited. 

https://doi.org/10.1016/j.jma.2021.07.019

https://doi.org/10.1016/j.biomaterials.2020.120477

Author Response

Dear reviewer, we would like to thank you for your valuable reviews and comments, which, we believe, helped us to improve our manuscript significantly. Our amendments with regard to your requests, including the links to relevant pages, are are in the attached file. Please see the attachment.

Reviewer 2 Report

Manuscript polymers-1594705 is well written and deserves publication. The main question addressed is the degradation of polydioxanone biliary stents and validation of Raman spectroscopy quantification degradation method. The paper is relevant and interesting and the topic is original. TG, DSC, FTIR, SEM, FT-Raman, mechanical tests and statistical analysis were performed to evaluate in vitro the hydrolytic degradation of ELLA-CS stents. Conclusions are consistent with the arguments presented and address the main question posed. It was find that the changes in mechanical properties are related to the increase in the ratio of amorphous to crystalline phase. The strength of the study is its novelty: the comparison of Raman data with the mechanical properties of a ppdx monofilament.

Few suggestion to improve the manuscript:

  1. Abstract : line 30: define meaning of “proper material degradation characteristics”

Line 33: Raman peak area (surface under the peak) cannot be in cm-1 (remove unit). The same through the manuscript (in table 2, line 487, line 516, etc. Intensity in Raman spectra has a.u.

  1. Fig 2: please add a scale bar if possible
  2. Line 129: please provide a picture of the seven samples cut of a stent if available.
  3. Line 164: please indicate the FTIR spectroscopy acquisition technique: ATR, DRIFT, etc. (if ATR or micro ATR please indicate the crystal type).
  4. Line 174: what are the units of 1800 g mm-1 grating about? g comes from gram of gratings/grids? The Raman spectrometer is a dispersive or a FT one?
  5. The grid in Fig. 2 is 6x3 not 8x4.
  6. Line 275: did you weighed the stents after 3, 6, 9 weeks degradation? This questions relates to the statement of line 279: “the earlier assumption that water content in the material increases during degradation was not confirmed”. Also, water content may increase after longer degradation times. Do you plan to extend the stent degradation characteristics survey?
  7. In table 2 please indicate amorphicity data values also.
  8. Fig 7: the FTIR spectra of stents before and after 9w degradation looks different, especially in the region of C-H valence stretching area of about 2900 cm-1. A visible intensity modification appears. You may try to obtain the crystallinity data from FTIR ratio of the bands intensities at 1372 and 2900 cm-1. Please check reference Chirila, Laura, Alina Popescu, Mihalis Cutrubinis, Ioana Stanculescu, and Valentin Ioan Moise. "The influence of gamma irradiation on natural dyeing properties of cotton and flax fabrics." Radiation Physics and Chemistry 145 (2018): 97-103 and other results. It was even tried to correlate crystallinity with XRD data.
  9. Fig 9. should be added as an inset of Fig 8. What method of area calculation was used (relative, absolute)?
  10. Try to exploit better the results correlating by linear regression plots the amorphicity with the tensile strength and also the 1732 peak area with the tensile strength.

Author Response

(The authors gave the same response as above.)

Round 2

Reviewer 1 Report

.